# Nanoplatform for the Delivery of Topotecan in the Cancer Milieu: An Appraisal of its Therapeutic Efficacy

**DOI:** 10.3390/cancers15010065

**Published:** 2022-12-22

**Authors:** Mohammed Kanan Alshammari, Mohammed Khalid Alghazwni, Abrar Saleh Alharbi, Ghayda Ghazi Alqurashi, Mehnaz Kamal, Salman Rahim Alnufaie, Salem Sayer Alshammari, Bandar Ali Alshehri, Rami Hatem Tayeb, Rashad Jameel M. Bougeis, Alaa Adel Aljehani, Nawaf M. Alotaibi, Abida Abida, Mohd. Imran

**Affiliations:** 1Department of Clinical Pharmacy, King Fahad Medical City, Riyadh 12211, Saudi Arabia; 2Department of Pharmaceutical Care, Security Forces Hospital, Riyadh 11538, Saudi Arabia; 3Department of Pharmaceutical Sciences, Maternity and Children’s Hospital, Mecca 24246, Saudi Arabia; 4Department of Emergency, Primary Healthcare Center, Mecca 24211, Saudi Arabia; 5Department of Pharmaceutical Chemistry, College of Pharmacy, Prince Sattam Bin Abdulaziz University, Al-Kharj 11942, Saudi Arabia; 6Department of Infection Control, Riyadh Third Health Cluster, Riyadh 13223, Saudi Arabia; 7Department of Pharmaceutical Care, Al-Dawaa Medical Services, Jubail 35412, Saudi Arabia; 8Laboratory Department, King Abdulaziz University Hospital, Jeddah 21589, Saudi Arabia; 9Department of Clinical Pharmacy, Faculty of Pharmacy, Northern Border University, Rafha 91911, Saudi Arabia; 10Department of Pharmaceutical Chemistry, Faculty of Pharmacy, Northern Border University, Rafha 91911, Saudi Arabia

**Keywords:** cancer, topotecan, side effects, passive targeting, active targeting, combinatorial drug therapy, clinical studies, patent

## Abstract

**Simple Summary:**

Nanotechnology has been implemented in healthcare more and more over the past few decades, particularly in applications for more efficacious and safer targeted delivery, detection, and therapy. Topotecan-loaded nanocarrier systems have shown superior pharmacokinetics, biocompatibility, tumor-targeting ability, and stability compared to topotecan in its native form. Additionally, they play a key role in reducing systemic toxicity and battling drug resistance. These advantages enable the widespread use of nano-based systems in various applications. This article explores nanoenabled active and passive targeting strategies and combinatorial therapy employing topotecan to ameliorate various cancers, along with a glimpse of the clinical studies utilizing the said molecule.

**Abstract:**

Chemotherapy has been the predominant treatment modality for cancer patients, but its overall performance is still modest. Difficulty in penetration of tumor tissues, a toxic profile in high doses, multidrug resistance in an array of tumor types, and the differential architecture of tumor cells as they grow are some of the bottlenecks associated with the clinical usage of chemotherapeutics. Recent advances in tumor biology understanding and the emergence of novel targeted drug delivery tools leveraging various nanosystems offer hope for developing effective cancer treatments. Topotecan is a topoisomerase I inhibitor that stabilizes the transient TOPO I-DNA cleavable complex, leading to single-stranded breaks in DNA. Due to its novel mechanism of action, TOPO is reported to be active against various carcinomas, namely small cell lung cancer, cervical cancer, breast cancer, and ovarian cancer. Issues of cross-resistance with numerous drugs, rapid conversion to its inactive form in biological systems, appended adverse effects, and higher water solubility limit its therapeutic efficacy in clinical settings. Topotecan nanoformulations offer several benefits for enhancing the therapeutic action of this significant class of chemotherapeutics. The likelihood that the target cancer cells will be exposed to the chemotherapeutic drug while in the drug-sensitive s-phase is increased due to the slow and sustained release of the chemotherapeutic, which could provide for a sustained duration of exposure of the target cancer cells to the bioavailable drug and result in the desired therapeutic outcome. This article explores nanoenabled active and passive targeting strategies and combinatorial therapy employing topotecan to ameliorate various cancers, along with a glimpse of the clinical studies utilizing the said molecule.

## 1. Introduction

A set of diseases collectively referred to as “cancer” are those in which uncontrolled tissue growth manifests malignantly, leaving cells without their usual form and/or functionality [1]. These abnormal cells can infiltrate other bodily areas, multiply, undergo angiogenesis, trick the immune system, and cause life-threatening malignancies [2]. Timely and proper therapy is mandated to prevent the unchecked growth of tumors and their impact on cancer patients. Cancer treatment is challenging, nevertheless, partly because of the unchecked metastatic pathways in invasion, neovascularization, circulation, extravasation, and migration, as well as the tumor cells’ increasing resistance to chemotherapeutic drugs [3].

Radiation therapy, surgery, hyperthermia, and stem cell therapy are examples of non-pharmacologic treatments that can be used. Combinatorial techniques between two or more therapeutic choices can also be sought after [4,5]. Intrusiveness, low drug solubility, brief circulation time of chemotherapeutics, multidrug resistance, indiscriminate targeting, and off-target adverse effects are a few of each approach’s drawbacks [6]. As a result, despite the variety of approaches that are already accessible, there is still a need to design safer and more effective therapeutic approaches. Additionally, advances in diagnosis and imaging methods would enable quicker cancer detection and better tracking of the disease’s trajectory, enabling us to comprehend the condition’s progression and tailor the appropriate treatment modalities accordingly.

Nanomedicine has emerged as a practical interdisciplinary approach for better managing human well-being, including the search for efficient anticancer therapies, to bridge this gap [7,8]. Nanotechnology offers a wide range of advantages by offering the conventional paradigms better localized therapeutic efficacy, less systemic toxicity, improved diagnostic sensitivity, and enhanced imaging capabilities [9,10,11,12,13]. It is recognized as a workable drug delivery channel for chemotherapeutic entrapped nanoparticle-based therapy due to its capacity to circumvent multidrug resistance and encapsulate and shuttle molecules with different physicochemical attributes to the desired tumor site [14]. Experts worldwide focus on the field mentioned above owing to the multiple benefits nanoformulations give over conventional therapeutics. For the targeted distribution of drugs for a multitude of therapeutic applications, a variety of nanocarriers have been used, including polymeric nanoparticles, nanoemulsions, liposomes, nanostructured lipid carriers, solid lipid nanoparticles, metallic nanoparticles, nano drug conjugates, dendrimers, hydrogels, carbon nanotube, and many others (Figure 1) [15,16,17,18].

Developing bioactive compounds that can be conjugated or encapsulated and released when exposed to the tumor microenvironment’s specific chemical and biological circumstances has received particular attention in research investigations. These approaches are termed passive and active targeting strategies, enabling the suitable delivery of chemotherapeutics at the targeted tumor site (Figure 2). These nanocarriers can increase a drug’s bioavailability, aggregate in a tumor effectively, stimulate tumor cell internalization, couple therapeutic compounds with imaging tools, and enhance antitumor activity [19].

Various research demonstrates topotecan’s anticancer properties, employing various nanoparticle-based delivery systems among the several camptothecin derivatives [21]. Topotecan is a topoisomerase 1 (TOP1) inhibitor. The chemical structure of topotecan is depicted in Figure 3. The vital enzyme TOP1 is found in higher eukaryotes and is crucial for DNA replication. DNA, a supercoiled double helix, untangles during the replication process to produce single strands that serve as a framework for synthesizing novel strands. When the helix starts to unravel, TOP1 joins DNA in a momentary cleavable complex and creates nicks in the DNA to reduce torsional stress. The enzyme is released owing to a transitory complex, enabling the new strand to be re-ligated [22].

Topotecan blocks the re-ligation of these single-strand breaks by attaching to the topoisomerase I-DNA complex. Due to the ternary complex’s interference with the replication fork in motion, DNA double-strand breakage and replication arrest are induced. The creation of this ternary complex finally results in apoptosis because mammalian cells cannot repair these double-strand breaks effectively. Topotecan stabilizes the TOP1 nicked DNA complex by binding and preventing the nicked strand from being re-ligated. As a result, irreversible double-strand breaks are induced to develop [23,24]. Topotecan and other camptothecin analogs’ lethal effects are largely produced while the tumor cells are in the cell cycle’s s-phase (DNA synthesis). Cells in the s-phase are 100 and 1000 times more responsive to camptothecins than in the G1 or G2 phases, according to in vitro research studies [25]. The hematologic adverse reactions associated with the drug use include neutropenia, anemia, and thrombocytopenia, while the non-hematologic adverse reactions were noted as nausea, diarrhea, vomiting, fatigue, and alopecia [25,26].

Topotecan has a roughly 3-h serum half-life, a substantial volume of distribution with significant tissue absorption, and minimal protein binding. A lactone ring serves as the basis for the chemical structure. Reversible hydrolysis converts topotecan from its biologically active lactone form to the inert open-ring carboxylate form. Additionally, it can cross the intact blood–brain barrier. Since the kidneys are responsible for most of the agent’s excretion, dose modification is required when renal function is compromised. On the other hand, patients with impaired hepatic function exhibit unchanged pharmacokinetic behavior [27].

Topotecan has a strong anticancer impact against breast cancer [28], ovarian cancer [29], cervical cancer [30], as well as small cell lung cancer [31], but its widespread use may be constrained by the degradation of its lactone ring (active form) to the carboxylate group (inactive form), which lowers its therapeutic effectiveness [32]. The occurrence of inactive form is noted to be more in these circumstances since this conversion takes place at the physiological pH. As a result, numerous attempts have been made to use a nanotechnology-based strategy to increase topotecan’s stability at a physiological pH, while retaining its anticancer benefits [14].

This paper seeks to thoroughly explain current developments in tumor-targeted treatment strategies using the chemotherapeutic topotecan. An in-depth discussion of recently developed nanosystems encasing topotecan that employ either passive or active targeted techniques for tumor amelioration is presented in this paper. More effort has been made to explain the benefits of investigating combinatorial drug therapy, which uses two or more chemotherapeutics to have a synergistic effect. The closing section of this paper presents an assortment of completed or continuing clinical trials undertaken to administer topotecan effectively, using various nanotools for cancer treatment.

## 2. Passive Targeted Delivery Approach for Topotecan

The purpose of passive targeting is to take advantage of the differences between tumor and normal tissues. Chemotherapeutics are efficiently transported to the target site by passive targeting to accomplish a therapeutic function. Substantial cancer cell multiplication causes neovascularization and wide fenestrations in the vascular wall, enhancing the tumor vessels’ permeability relative to healthy vessels [33]. Macromolecules, such as NPs, might escape from blood arteries supplying the tumor and amass within tumor tissue due to the fast and deficient angiogenesis. The accumulation of NPs is increased in cancer due to inadequate lymphatic drainage, which enables the nanocarriers to transfer their payloads to tumor cells. These procedures result in the EPR effect, which is one of the drivers behind the passive targeting approach [34]. In conjunction with the EPR effect, the tumor milieu plays a significant role in the passive distribution of nanomedicines. One of the metabolic traits of cancer cells is glycolysis, which serves as the primary energy supply for the development of cancer cells [16]. The tumor microenvironment’s pH is decreased by glycolysis, which creates an acidic setting. As a result, some pH-sensitive NPs are activated by the lower pH and can release medications close to cancer cells [35]. Based on the set premise, various topotecan nanoparticles were targeted to the tumor tissues to confer superior therapeutic effectiveness.

Topotecan (TOPO) was encased in mesoporous silica nanoparticles (MSNs), and the nanosystem allowed the drug’s active form to be delivered in endosomes/lysosomes (pH 5.5) upon the internalization of nanoparticles. A pH-sensitive coating, a multimodal gelatin shell that protected TOPO from hydrolysis and premature release, and several anchorage sites for marking targeted ligands for preferential uptake in tumor cells were the hallmarks of MSNs. The nanosystems effectively destroy tumor cells while not affecting normal cells’ survival. On the other hand, free TOPO could not kill both cell lines due to the drug’s deactivation. This revolutionary nanodevice represents a step ahead in developing new cancer-fighting weaponry [36].

A topotecan-entrapped liposomal nanoformulation (LNP) was developed based on a loading process that entails the production of a copper water-soluble camptothecin complex. The same loading process developed for irinotecan was followed to produce an LNP topotecan formulation (Topophore C). At a final drug-to-lipid (D/L) mole ratio of 0.1, the entrapment efficiency of topotecan was noted to be >98%. Greater D/L ratios were possible; however, in vitro drug release tests revealed that the ensuing formulations were less stable. Topotecan plasma half-life and AUC were raised 10- to 22-fold in mice after Topophore C treatment, compared to free topotecan. Topophore C was noted to be 2-to 3-fold more toxic than free topotecan, but it had considerably superior anti-tumor effectiveness with no adverse effects. Based on the inferred findings, it can be inferred that Topophore C is a promising pharmacological candidate for treating platinum-resistant ovarian cancer [37].

Topotecan would continue to benefit from the targeted site delivery by utilizing nanocarriers. Anti-epidermal growth factor receptor (EGFR) and anti-human epidermal growth receptor 2 (HER2)-immunoliposome formulations substantially boosted topotecan internalization compared to the non-targeted counterparts and free topotecan, resulting in enhanced cytotoxic activity and superior antitumor efficacy against HER2-overexpressing human breast cancer (BT474) xenografts. Topotecan’s targeting capability and pharmacokinetic properties were considerably improved when it was stabilized in nanoliposomes, enabling potent and effective formulations against solid tumors [38].

Topotecan was further reported to have enhanced efficacy in ovarian cancer. An appreciable particle size and entrapment efficiency of 60.9 ± 2.2% were obtained in a nanometric range. The formulated nanoparticles illustrated a sustained release in physiological and acidic tumor microenvironmental conditions. The nanometric size enabled ideal internalization in SKOV3 (ovarian cancer) cell lines over time compared to topotecan in a soluble form, with a 13.05-fold rise in bioavailability when evaluated for pharmacokinetic attributes [39].

The polylactic-co-glycolic acid (PLGA) nanocarrier was also used to formulate topotecan (TPT), which improved the drug’s efficacy by reducing the accelerated conversion of the bioactive lactone form to the inactive carboxylate form. TPT’s stability was ensured by maintaining the drug-containing phase at an acidic pH. The drug maintained its active lactone form by lowering the pH of the inside of nanoparticles, which led to a 15-day biphasic release profile. Furthermore, compared to a neat drug, the cytotoxicity screening and in vivo antitumor effectiveness revealed considerable potential for greater proliferation inhibition [14].

Nanostructured lipid carriers (NLC) incorporating topotecan (TPT-NLC) were fabricated in hydrogels with hydroxyethyl cellulose and chitosan (TPT-NLC-HEC and TPT-NLC-Ch). For around 30 days, the said formulations retained the drug and nanoparticle dispersions stably. TPT release was dramatically reduced when nanoparticles were added to gels. TPT-NLC-HEC boosted permeability by 2.37 times compared to TPT-HEC (11.9 and 5.0 g/cm^2^, respectively). Nanoencapsulation significantly increased TPT cytotoxicity when analyzed in B16F10 melanoma cells. With an IC50 value of 5.74 g/mL, TPT-NLC was noted to be more toxic than free TPT, whereas free TPT had an IC50 of >20 g/mL. Because the skin penetrated values of TPT from the established formulation (TPT-NLC) were higher than the melanoma IC50, it may be stated that chemotherapeutic permeated quantities may be adequate for a therapeutic impact [40].

TPT-SLNs were integrated into a thermoresponsive hydrogel system (TRHS) to create TPT-SLNs-TRHS, which allowed for controlled drug release and reduced drug-associated toxicity. When TPT-SLNs-TRHS was injected into the rat’s rectum, it showed good gelation capabilities. Furthermore, drug release was demonstrated to be controlled over an extended period for the integrated TPT. TPT bioavailability was improved with enhanced plasma concentration and area under the curve (AUC) in pharmacokinetic investigations. Furthermore, compared to the test formulations, it significantly improved antitumor impact in tumor-bearing animals. The study inferred that SLNs combined with TRHS could be a potential source of antitumor drug delivery with improved control over drug release, with no associated toxicity [41].

To augment topotecan’s transport to the lymphatic system, a primary conduit for cancer metastasis, and further enhance topotecan’s bioavailability and retention in target organs such as lung and brain, a research group formulated topotecan-loaded polymeric nanoparticles [42]. The cumulative percentage of topotecan release from the nanoformulation after a time period of 120 h were 91.56 and 92.02%, respectively, according to the results of in vitro release assays for the nanoformulation and free drug as a reference standard. PLGA nanoparticles with topotecan loading displayed a protracted release pattern in the studied time frame. Following 6 h after treatment, topotecan distribution was noted to be larger in each target organ after administration of the topotecan nanoformulation at a dose of 4 mg/kg than following delivery of the free drug. A similar pattern was seen following oral administration. The substantial intensity of luminescence was demonstrated for six hours following the injection of the nanoformulation. Higher luminescence in lymphoid tissues was noted, which was coherent with the quantitative observation of significant topotecan in the said tissues after intravenous administration of these nanoparticles. The outcomes of this study imply that topotecan NPs may produce superior therapeutic outcomes because they have a better pharmacokinetic profile and are efficient enough to distribute the drug more effectively to lymphoid tissues, the lung, and the brain as contrasted to the free drug [42].

The intrusion of rest intervals between cycles of chemotherapy became requisite with the advent of the “maximum tolerated dose” in standard treatment protocols resulting in direct toxicity. A novel drug delivery method called “metronomic chemotherapy” has been suggested to circumvent the issues brought on by conventional chemotherapeutic regimens [43]. Chemoresistance and toxic side effects are significant obstacles in chemotherapy, and metronomic chemotherapy has arisen to overcome these issues. The treatment plan targets activated endothelium cells in tumors by often administering standard chemotherapeutic drugs at extremely low doses; benefits include minimal side effects and a low chance of acquiring drug resistance. Before the recent discovery of other processes, it was believed they acted by targeting angiogenesis; nevertheless, metronomic chemotherapy has now been recognized as a multitargeted therapy. Better cancer treatment regimens will be created using clinical experience and the knowledge gathered from metronomic chemotherapy’s preclinical investigations [44]. Low-dose metronomic chemotherapy (LDMC) is the term used to describe the sustained administration of low-dosage chemotherapeutics to reduce toxicity and specifically target the tumor endothelium to prevent tumor proliferation. When topotecan (TPT) is used at its maximum tolerated dosage (MTD), systemic hematological complications are frequently reported. By protecting TPT from circulatory clearance and enabling higher absorption and prolonged tissue exposure in tumors, nanoencapsulation of TPT has shown improved effectiveness.

Additionally, nanoformulations with extended TPT release can imitate metronomic therapy with fewer doses. In clinical studies and preclinical carcinoma models that evaluated numerous cancer types, metronomic TPT has been shown to have favorable results [45]. The said study inferred that when measured over a long time, low quantities of topotecan entrapped in liposomal formulation and radiation increase the cytotoxicity of tumor-endothelial spheroids.

According to some reports, metronomic therapy using oral topotecan may show promise for use in metastatic colon cancer clinical trials [46].

## 3. Active Targeting

When the nanocarriers reach the tumoral zone, they face a challenging situation. Tumoral aggregates comprise various cell types, ranging from tumoral cells to immunological, supporting, and healthy cells from the extracellular matrix [47]. Consequently, nanocarriers must be able to distinguish malignant cells and localize their action on them to accomplish an effective therapeutic effect. By binding targeting moieties to the particle surface, this capability can effectively be included in the nanocarriers [48]. These targeting components are small compounds or macromolecules that engage specific receptors on tumor cells’ surfaces. These cellular receptors are found in healthy cells in many cases, such as for the extensively used targeting moieties folic acid [49], transferrin [38], and sugars [50]. However, because tumor cells have a larger nutrition need, their population is much higher than healthy cells. As a result, this receptor overexpression can be leveraged to selectively deliver therapeutic medications to tumor cells. Another option is to produce synthetic targeting components that are more selective and efficacious at binding to specific receptors [51].

Mesoporous silica nanoparticles laden with topotecan were synthesized and then surface-conjugated with folic acid (FTMN) to increase the drug’s effectiveness in treating retinoblastoma (RB) cancers. In physiological settings, the particles were nanosized and showed a controlled release of the entrapped drug. Compared to non-targeted nanoparticles, the folic acid-conjugated nanoformulations had a phenomenal absorption in RB cells. These findings strongly suggest that cellular uptake was regulated by receptor-mediated endocytosis. Compared to other formulations, FTMN had a considerably larger cytotoxic effect in Y79 cancer cells due to its higher cellular absorption. FTMN successfully triggered cancer cell death with a 58% effectiveness. The anticancer efficiency of TPT nanoformulation in Y79 cancer cells was superior to that of native drugs or unconjugated nanoparticles, according to the findings. FTMN revealed a decrease in overall tumor volume compared to the other group and fewer tumor cells in histological staining. Consequently, a folic acid-conjugated nanocarrier system could be a promising therapy option for RB [52].

Transferrin-decorated multifunctional nanoparticles (NPs) based on γ-cyclodextrin for tumor-targeted therapy were reported. The formulated NPs were found to cause a considerable increase in cellular uptake in MDA-MB-231 tumor cells leading to cell death. The transferrin-targeted NPs were proven effective carriers of TPT in vivo experiments using an MDA-MB-231 tumor xenografted mice model. TPT has the preferential ability to deliver chemotherapeutics to Tf receptor-positive MDA-MB-231 tumor cells, increasing drug uptake into the tumor cells and intensifying their toxicity [53].

Irrespective of the availability of numerous nanocarriers, researchers worldwide have explored liposomes as a suitable carrier system for the targeted delivery of topotecan employing specific ligands. Table 1 presents the list of studies about the targeted delivery of topotecan in the liposomal carrier system.

## 4. Combinatorial Drug Therapy Employing Topotecan

One of the cornerstones of cancer therapy is combinatorial therapy, which utilizes two or more chemotherapeutics to target cancer-inducing or cell-sustaining mechanisms [56] selectively. Even though the mono-therapy technique is still a prevalent type of cancer treatment, it is typically thought to be less efficient than the combination therapy strategy. Typical mono-therapeutic approaches non-selectively target actively multiplying cells, inevitably resulting in the death of both malignant and healthy cells. Chemotherapy can harm the patient and comes with several hazards and side effects. It often leads to drug resistance and cancer cell survival (Figure 4). It can also significantly lower the patient’s immune system, by weakening bone marrow cells and making them more vulnerable to host illnesses. The toxicity aspect of combinatorial therapy is greatly reduced because diverse channels will be targeted, even if it can still be harmful if one of the medications is a chemotherapeutic. Combinatorial therapy has a synergistic effect, necessitating a reduced therapeutic dosage of each chemotherapeutic separately [57]. Combinatorial therapy may also provide cytotoxic effects on cancer cells while preventing harmful effects on healthy cells.

In a drug-sensitive cancer cell, a strong drug-target interaction occurs, followed by the internalization of the anticancer drug at the target site leading to cell death. In contrast, in a drug-resistant cancer cell, a weak/inactive drug-target interaction occurs, followed by the poor internalization of the anticancer drug at the target site leading to cell survival.

Due to limited drug diffusion across the blood–brain barrier (BBB), the prevalence of MDR, and inadequate uptake into tumor tissues, chemotherapy for brain malignancies continues to be a challenge. Tamoxifen was integrated into the liposomes, and wheat germ agglutinin was coupled to the surface of liposomes. Topotecan was further loaded into the preformulated liposomes. The ligand-modified topotecan liposomes displayed a considerable inhibitory effect in the MTT experiment compared to the unconjugated formulation, implying that both the ligands confer robust drug delivery benefits into brain tumor cells following immediate drug exposure. The lowering of C6 glioma tumor spheroid volume and apoptosis was also noted. The combinatorial effects were observed in brain tumor-bearing rats, culminating in a considerable enhancement in the overall survival of the treated rats.

Furthermore, data from an extended treatment group showed that survival could be improved, implying that protracted chemotherapy with topotecan liposomes modified with TAM and WGA would be favorable for the efficacious treatment. To summarize, targeted topotecan liposomes enhance topotecan trafficking across the BBB, demonstrating dual-targeting benefits. These results could pave the way for more noninvasive brain tumor treatments [59].

A combinatorial therapy involving paclitaxel (Pac) and topotecan (Top) in a Pac-Top ratio of 20:1 w/w was incorporated into folate-anchored PEGylated liposomes (FPL-Pac-Top) for effective ovarian cancer treatment was reported by a group of researchers. Toxicity to blood cells was found to be minimal in hematological experiments. Long circulatory behavior and preferential accumulation of FPL-Pac-Top in the ovaries were observed in vivo. In addition, FPL-Pac-Top revealed lower necrosis and greater apoptosis. Compared to the Pac-Top solution, Kaplan–Meier survival analysis demonstrated a two-fold increase in the survival time by FPL-Pac-Top. The potential efficacy of FPL-Pac-Top was related to some characteristics, including thermosensitivity, extended circulatory nature, and targetability. Such an envisaged method could be a revolutionary chemotherapeutic technique for safe and effective cancer-targeting options [60].

The new purine derivative F7, which targets the abnormal disequilibrium between apoptosis and cell multiplication and exhibits broad-spectrum anti-tumor actions, is a prospective anticancer medication. F7 has demonstrated therapeutic benefits in the lung, breast, and hepatic cancer, among other malignancies. The introduction of F7 as a drug was hampered by evidence of its significant systemic toxicity in mice. Thus, to lower dosage and adverse effects while preventing drug resistance, F7 therapies had to be combined with other medications. A co-loaded thermosensitive liposome (F7-TPT-TSL) was developed to stabilize the drug in the inner aqueous phase of the formulation [61]. The dummy liposomes and co-loaded liposomes were stable in water and serum after being stored at 4 °C for a month, showing that the liposomal formulation had great physical stability and exhibited no signs of agglomeration. Superior cytotoxicity was conferred by the co-loaded formulation when compared to the single drug formulation (*p*  <  0.05), especially after heating (*p*  <  0.05). This may be attributed to the fact that heating at a temperature of 41 °C enhanced the cell membrane permeability, rendering the cells more inclined to internalize liposomes. A high temperature accelerated the cells’ accumulation of F7 and TPT, which subsequently killed the cells by synergistically acting on many targets. When tested in a xenograft tumor model created by injecting MCF-7 subcutaneously, the co-loaded liposome formulation showcased a higher tumor suppression rate than the free drug. Hence it may be inferred that the co-loaded liposomes could be a prospective therapy for temperature-triggered cancer treatment. 

To increase the penetration of topotecan through the gut and breast cells to treat breast cancer, tamoxifen citrate was added to the formulation to generate dual drug-loaded nanoparticles [62]. Tamoxifen is a P-glycoprotein (P-gp) inhibitor that helps decrease the drug’s adverse effects by lowering its dose. It was observed that the optimized dual drug-loaded nanoparticles had a smooth and spherical architecture. The in vitro release study demonstrated the sustained release of both the entrapped drugs. The ex vivo gut permeation investigation showed that TAM improved TOP’s penetration and raised its bioavailability by 1.9-fold. Cell line experiments on MCF-7 cells were conducted further to establish the penetration and cytotoxicity of the drug combination. Comparing the dual drug-loaded nanocarrier system to native drugs alone and in combination, it was found to be significantly more lethal at low concentrations. The P-gp inhibition caused by tamoxifen on the cell surface and the nanoparticles’ relatively small size, making it easier for them to enter cells, are most likely responsible for the said effect.

Additionally, compared to pure drugs or drug mixtures, dual drug-loaded nanoparticles demonstrated lower IC50 values leading to superior cell death. The reason could be that tamoxifen inhibits P-gp; it helps restrict topotecan from effluxing off breast cancer cells (MCF-7). The obtained results implied the superior cytotoxic potential of the system in facilitating breast cancer. Hence, it is imperative to state that combinatorial therapy was more efficacious in treating breast cancer than single therapy.

## 5. Clinical Studies of Topotecan Nanoformulations

Even though thousands of pre-clinical studies on topotecan-based nanoformulations have already been published, not many formulations have reached the clinical stage. This may result from the formulations’ inconsistent results when manufactured on a larger scale or the therapeutic effects they exhibit when tested in human research.

Although none of the topotecan nanoformulations has yet entered the market, some of them are now through various stages of clinical testing, which upon conferring promising results, may hit the commercial front (Table 2). The presented data are extracted from the site “https://www.clinicaltrials.gov/”(accessed on 20 November 2022) upon entering the keywords “topotecan nanoformulation.”

## 6. Patent Literature

The patent literature search revealed some important patents and applications related to TOPO nanoformulations (Table 3). The nanoformulation-based invention of TOPO has been patented (Table 3).

Almost all the patented inventions of TOPO were directed to provide a biocompatible, quality nanoformulation of TOPO with improved delivery of TOPO to the tumor cells, enhanced curative effect, and reduced toxicity of TOPO. Some nanoformulations-based inventions of TOPO also claim to reverse the multidrug resistance of tumor cells [63,64] and avoid degradation of the TOPO [65]. All cited patents/patent applications in Table 3 also provide the process for preparing the claimed invention.

## 7. Authors’ Opinion/Analysis

Over the past couple of decades, targeted delivery methods for anticancer medications have dramatically increased. Despite many promising preclinical trials, only a limited fraction of passively targeted nanocarriers and none of the actively targeted nanoparticles have received clinical approval. Hence, a proper selection of nanocarrier systems for the chosen drugs needs to be made and optimized intelligently. The traits of the tumour cells and the therapy’s chemical makeup should be considered before choosing between active or passive tumor targeting. An “intelligent” theranostics platform must be established by combining an active or passive drug carrier with an imaging or diagnostic agent. This system will be able to track the evolution of the dreadful disease and assess the drug’s therapeutic efficacy promptly. Such systems necessitate a careful investigation of tumor biology as well as the investigation of new target possibilities and unique drug carriers. The development of nanocarriers with tunable biological identities is required to hasten the clinical translation shortly.

Numerous clinical trials have been conducted, many patents have been issued in the last 20 years, and some nanocarriers have even made it to the patients’ bedsides. However, it is crucial to gather evidence on the safety profile of different excipients employed in creating nanocarriers and the final formulations. This is a crucial factor in assessing a formulation’s safety before using it to treat a specific illness. When tested in vivo, the individual components of nanoparticulate delivery systems degrade in various ways that could result in little to low toxicity. Since the excipients utilized in the delivery system significantly affect the system’s overall effectiveness and safety, a comprehensive examination of the nanoparticle-based drug delivery system necessitates a detailed grasp of the composition of nanoparticles. Establishing the potential risk of the excipients and the optimized formulation is, therefore, a difficult endeavor because it calls for extensive long-term research, which also demands a proper understanding of toxicology guidelines relevant to the route and dose of in vivo administration, the manufacturing procedure and storage circumstances, as well as the biological fate of the drug.

Additionally, stricter regulations regarding the security of the excipients used in nanoformulations must be implemented. More in-depth research is necessary to investigate the general safety of delivery systems based on nanoparticles. A safe and effective nanomedicine will be possible, thanks to the development of a nanoformulation and the intricacies of in vivo safety profiles of the excipients employed in nanocarriers.

An effective nanoparticulate-based delivery system’s future will primarily depend on the formulation’s safety, determined by its constituent parts’ toxicological consequences.

## 8. Conclusions and Future Perspectives

Nanotechnology has been implemented in healthcare more and more over the past few decades, particularly in applications for more efficacious and safer targeted delivery, detection, and therapy. Topotecan-entrapped nanocarrier systems have shown superior pharmacokinetics, biocompatibility, tumor-targeting ability, and stability compared to topotecan in its native form. Additionally, they play a key role in reducing systemic toxicity and battling drug resistance. These advantages enable the widespread use of nano-based systems in various applications. Many significant obstacles remain to overcome before anticancer nanomedicines are clinically translated. For instance, some relevant queries include how to improve patient population categorization in clinical trials, how to improve the dosage schedule of nanomedicines in combinatorial therapies, and how to guarantee excellent performance and consistency for industrialized nanomedicine manufacturing that surface up in relation to the clinical translation of the nanomedicines. The basic technology of related research and development is anticipated to be defeated by the integration of molecular-level scientific modeling and exquisite control of process engineering with the advancement of nanotechnology research, creating a new arena for novel drug delivery systems.

Topotecan encapsulated in nanocarriers also provides improved opportunities for combinatorial therapy, which helps overcome drug resistance mechanisms such as efflux transporter upregulation, an improper apoptotic pathway, and a hypoxic tumor milieu. According to various MDR mechanisms, NPs laden with various ligands and chemotherapeutics like topotecan can reverse the effects of drug resistance. Even though many passive targeted therapies have been explored extensively, a lesser number of researches have been reported related to the active targeting of topotecan nanoformulations. Research approaches for the targeted delivery of topotecan nanoparticles will evolve more with time as a result of research investigations on the biological traits of specific tumors. Therefore, it is essential to note that improved knowledge of the tumor microenvironment and additional research into how nano-based drug delivery systems interact are necessary to develop novel targeted delivery systems.

## Figures and Tables

**Figure 1 cancers-15-00065-f001:**
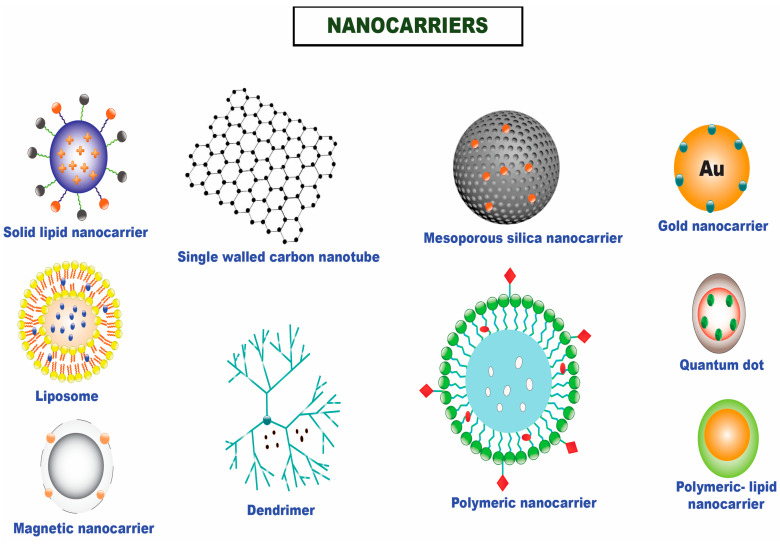
Different nanocarriers that are utilized for the efficient delivery of chemotherapeutics in solid tumors.

**Figure 2 cancers-15-00065-f002:**
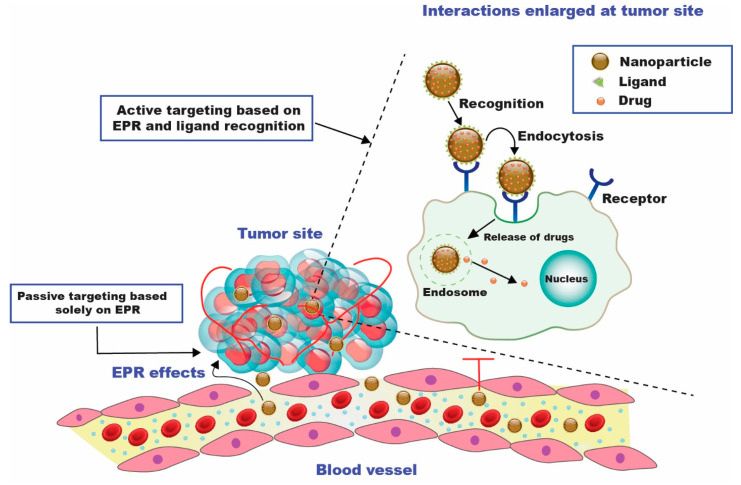
Diagrammatic representation of passive and active drug targeting techniques. For passive targeting, the enhanced permeability and retention (EPR) effect causes the nanocarriers to aggregate at the tumor site after passing through the leaky walls. Active targeting can be accomplished by utilizing certain ligands specific to the over-expressed receptors on the tumor cells [20].

**Figure 3 cancers-15-00065-f003:**
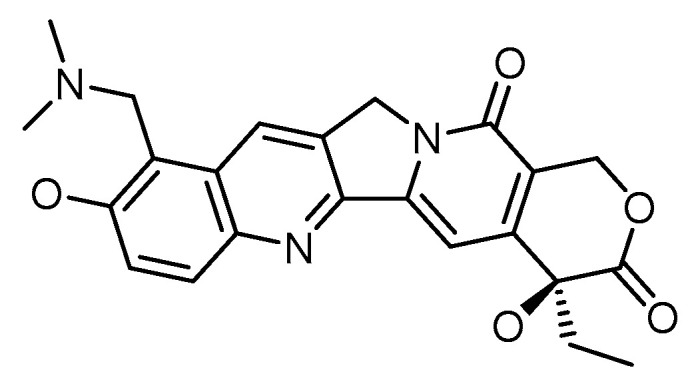
Chemical structure of topotecan.

**Figure 4 cancers-15-00065-f004:**
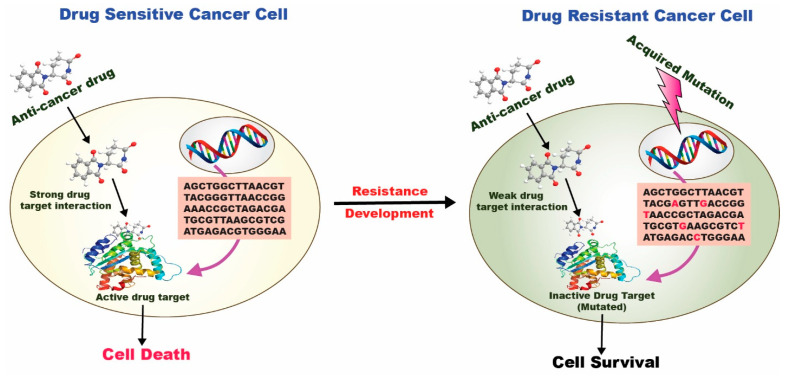
Drug resistance mechanism in a drug-sensitive and drug-resistant cancer cell [58].

**Table 1 cancers-15-00065-t001:** List of studies about the targeted delivery of topotecan in the liposomal carrier system.

Nanocarrier	Targeting Ligand	Cell Line/In Vivo Model	Outcomes	Ref
Liposome	scFv F5, C225 Fab	HER2 positive, EGFR positive cell line/NCR nu/nu athymic female mice	Targeted formulations significantly boosted topotecan internalization with a concomitant rise in cytotoxicity and enhanced antitumor efficacy.	[40]
Liposomes	Dequalinium (DQA)	MCF7 and MCF7/ADR cells/MCF7/ADR tumor-bearing mice	A 2-fold and 4-fold-higher cytotoxic potential was conferred by the targeted ones than free topotecan. According to the co-localization study, the liposomes carried the payload to mitochondria at 21.2-fold higher than free form and 12.9 factors higher than non-targeted liposomes. Compared to the other treatment groups, the DQA-targeted liposomes significantly inhibited tumor growth, according to an antitumor investigation in mice bearing the MCF7/ADR tumor.	[28]
Liposome	Folic acid	A549 cells/xenografted A549 cancer cells in female nude mice	Targeting lung cancers with folate-targeted topotecan liposomes reduced tumor volume and prolonged blood circulation duration.	[54]
Liposome	Polyethylene glycol (PEG)	_	Around 100% loading efficiency was attained. Compared to the free drug, the PEG-coated liposomes were cleared from blood circulation at a comparatively slower pace. Additionally, compared to normal tissues, tumor tissues accumulated more nanoliposomal formulation than normal cells did with topotecan in its free form under the same circumstances.	[55]

**Table 2 cancers-15-00065-t002:** List of clinical studies employing targeted topotecan nanoformulations.

Malignancy	Clinical Trial Phase	Clinical Trial Number	Intervention/Treatment	Status	
Advanced Solid Tumors	I	NCT04047251	Drug: FF-10850 Topotecan Liposome Injection	Recruiting	Advanced Solid Tumors
Small Cell Lung Cancer, Ovarian Cancer	I	NCT00765973	Drug: Topotecan Liposomes Injection (TLI)	Completed	Small Cell Lung Cancer, Ovarian Cancer

**Table 3 cancers-15-00065-t003:** Important granted patents/patent applications of TOPO.

Patent/Application Number (Applicant)	Summary
**CN102697735A** (Nanjing University)	This Chinese patent application claims a biocompatible and biodegradable polymer-based nanoformulation of TOPO [63].
**CN102764234A** (Shanghai Modern Pharmaceutical)	This Chinese patent application claims topotecan hydrochloride targeting liposome modified with a hydrophilic polymer polyethylene glycol (PEG) and a targeting ligand RGD peptide [64].
**US2006222694A1** (SmithKline)	This US patent application claims a lyophilized topotecan liposomal composition comprising topotecan, liposomes and cryoprotectant [65].
**US9295735B2** (Medgenesis Therapeutix)	This US patent claims a therapeutic composition comprising a non-PEGylated liposomal delivery vehicle for administering TOPO via convection-enhanced delivery to the central nervous system [66].
**US7060828B2** (Inex Pharmaceuticals)	This US patent claims a liposomal topotecan unit dosage form comprising a mixture of lipid (sphingomyelin and cholesterol) and topotecan [67].
**CN109999009B** (Chongqing Medical College)	This Chinese patent claims an emulsification method based on oral sustained-release microsphere preparation of TOPO [68].
**CN104771361B** (Chinese Academy of Sciences)	This Chinese patent claims a phospholipid-based topotecan hydrochloride liposome nanoformulation of TOPO [69].
**CN102716085B** (Hainan Lingkang Pharmaceutical)	This Chinese patent claims a cholesterol succinate-based topotecan hydrochloride liposome injection [70].
**CN101744767B** (Academy of Military Medical Sciences)	This Chinese patent claims a common lipids-based thermosensitive liposome preparation containing TOPO [71].
**CN102429870A** (China Pharmaceutical University)	This Chinese patent application claims a polyamide-amine dendrimer-based targeting nanocarrier for carrying TOPO [72].
**CN101744764A** (Shanghai Institute of Pharmaceutical Industry)	This Chinese patent application claims an amphiphilic lipid-based topotecan hydrochloride polycystic liposome [73].
**CN101015526A** (Jiangsu Aosaikang Pharmaceutical)	This Chinese patent application claims a biologically acceptable phospholipid-based topotecan liposome [74].
**CN103479568A** (China Pharmaceutical University)	This Chinese patent application claims a lipid and situ-gel substrate-based topotecan hydrochloride intratumor injection liposome preparation [75].

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
