# Peer review of "Nanoplatform for the Delivery of Topotecan in the Cancer Milieu: An Appraisal of its Therapeutic Efficacy"

_cancers, 2022, doi:10.3390/cancers15010065_

Round 1

Reviewer 1 Report

This review describes chemotherapeutics of topotecan loaded nanocarriers. This further describes passive and active targeted techniques for tumor amelioration followed by combinatorial drug therapy. A few strategies used in with topotecan which are in the research stage, clinical trials and patents are
mentioned. The compilation of research and strategies used specifically for cancer is beneficial to the research community. The word ‘appraisal’ mentioned in the title of the manuscript should be justified by including a few paragraphs of explorative comments, without which the content will simply be a compilation of research summary. Therefore, a section on authors’ perspectives on these strategies is expected in the review to discuss further about the appraisal.
The following are the comments about the review manuscript.
1. Nanoplatform: Although, nanoplatforms are mentioned in the compilation of research, a summary of nanocarrier strategies and authors’ analysis of these is missing in the review. Please include this.
2. Line 211 mentions Nanotoxicity. Line 252 systemic hematological complications. These are important aspects of nanoparticle toxicity which are an essential part of nanoparticle research. Please do elaborate about this attribute of nanoplatforms.
3. Line 247 It will be good to explain metronomic chemotherapy for the benefit of readers.
4. Appraisal of its therapeutic efficacy
4.1. The manuscript mentions these in the title. Therefore, it is essential to dedicate few paragraphs on these topics elucidating the authors’ insights to enlighten readers on these aspects.
4.2. Or a separate paragraph on authors’ perspectives about therapeutic efficacy of chemotherapeutics will be useful.

Author Response

Please refer the attached file

Reviewer 2 Report

The presented study is in the field of nanotechnology and health care. Nanotechnology is a promising branch of modern strategies in cancer therapy. Nanotechnologies provide the possibility of drug delivery in target tissues and even cells, for this reason the development in this area is not only of scientific interest, but also of applied. In the present review, the knowledge regarding the combined administration of topotecan-antitumor agent and possible nanoparticle options for its passive and active delivery is systematized.

I have the following remarks to the authors:

1. The sources of figures 1 and 3 are not indicated.

2. Please provide more details on the structure and mechanism of action and properties of topotecan.

3. Not all abbreviations are given in full text, which makes reading difficult, for example: AUC on line 178; PLGA on line 199; TPT on line 201 and others.

4. In the paragraph from lines 227 to 244, only one quote is indicated, are these the authors' own data or are they taken from literary sources? The same question applies to the paragraph from line 353 to line 374, as well as to the statement expressed on lines 380-387.

5. No reference sources were presented for some of the clinical studies (Table 2).

I believe that after removing these inaccuracies, the work could be published in a journal Cancers.

Author Response

Please refer the attached file

Round 2

Reviewer 1 Report

N/A

Reviewer 2 Report

I agree all corrections. Now my opinion is that MS is ready for publication.